# Clinical Course, Laboratory Findings, and Prognosis of SARS-CoV-2 Infection in Infants up to 90 Days of Age: A Single-Center Experience and a Proposal for a Management Pathway

**DOI:** 10.3390/healthcare12050528

**Published:** 2024-02-23

**Authors:** Tommaso Bellini, Giacomo Brisca, Ioannis Orfanos, Marcello Mariani, Federico Pezzotta, Benedetta Giordano, Andrea Pastorino, Silvia Misley, Clelia Formigoni, Elena Fueri, Marta Ferretti, Marta Marin, Martina Finetti, Emanuela Piccotti, Elio Castagnola, Andrea Moscatelli

**Affiliations:** 1Pediatric Emergency Room and Emergency Medicine Unit, Emergency Department, IRCCS Istituto Giannina Gaslini, 16147 Genoa, Italy; martaferretti@gaslini.org (M.F.); martamarin@gaslini.org (M.M.); martinafinetti@gaslini.org (M.F.);; 2Neonatal and Pediatric Intensive Care Unit, Emergency Department, IRCCS Istituto Giannina Gaslini, 16147 Genoa, Italy; giacomobrisca@gaslini.org (G.B.); andreamoscatelli@gaslini.org (A.M.); 3Department of Clinical Sciences, Lund University, 22100 Lund, Sweden; ioannis.orfanos@med.lu.se; 4Department of Pediatrics, Skåne University Hospital, 22185 Lund, Sweden; 5Infectious Diseases Unit and COVID Hospital, Department of Pediatrics, IRCCS Istituto Giannina Gaslini, 16147 Genoa, Italyeliocastagnola@gaslini.org (E.C.); 6Department of Neuroscience, Rehabilitation, Ophthalmology, Genetics, Maternal and Child Health (DINOGMI), University of Genoa, 16132 Genova, Italy; 5296194@studenti.unige.it (F.P.); 4043931@studenti.unige.it (B.G.); 5288131@studenti.unige.it (A.P.); 4008194@studenti.unige.it (S.M.); 3805091@studenti.unige.it (C.F.); 3764730@studenti.unige.it (E.F.)

**Keywords:** clinical severity, COVID-19 epidemiology, febrile infant, pediatric emergency department, prognosis

## Abstract

Aim: To provide a comprehensive description of the clinical features, biochemical characteristics, and outcomes of infants up to 90 days old with COVID-19. Moreover, to assess the severity of the disease and propose an effective management pathway. Methods: Retrospective single-center study spanning three years. Patient data includes age, sex, symptoms, comorbidities, blood and urine test results, cultures, admission, length of stay, therapies, intensive care unit admission, and mortality. Results: A total of 274 patients were enrolled in the study, comprising 55% males. Among them, 60 patients (22%) were under the age of 29 days, while 214 (78%) fell within the 29 to 90 days age range. The overall incidence of SARS-CoV-2 infections was 0.28 per 10,000 Pediatric Emergency Department admissions. Blood inflammatory markers showed no significant abnormalities, and there were no recorded instances of positive blood cultures. Less than 1% of infants showed urinary tract infections with positive urine cultures, and 1.5% of patients had a concurrent RSV infection. Hospitalization rates were 83% for neonates and 67% for infants, with a median length of stay (LOS) of 48 h for both age groups. None of the patients required admission to the Pediatric or Neonatal Intensive Care Unit, and only one required High Flow Nasal Cannula (HFNC). No secondary serious bacterial infections were observed, and all hospitalized patients were discharged without short-term sequelae. No deaths were reported. Discussion and Conclusions: Infants with COVID-19 generally exhibit milder or asymptomatic forms of the disease, making home management a viable option in most cases. Blood tests, indicative of a mild inflammatory response, are recommended primarily for children showing symptoms of illness. Hospitalization precautions for infants without apparent illness or comorbidities are deemed unnecessary. Given the evolving nature of experiences with COVID-19 in infants, maintaining a high level of clinical suspicion remains imperative.

## 1. Introduction

Coronavirus Disease-19 (COVID-19) is caused by Severe Acute Respiratory Syndrome-Coronavirus-2 (SARS-CoV-2) [1]. However, information about pediatric patients, especially those under 90 days of age, remains fragmented, scarce, and often contradictory [2,3,4]. Knowledge of these young patients has been extrapolated from older pediatric cases, with a limited number of research and case series publications during the ongoing pandemic [2].

The potential risk factors for community-acquired COVID-19 in very young individuals are unclear, with speculation about whether immature immune systems pose a risk or play a protective role [2,5,6,7,8]. Several risk stratification scores for COVID-19 in young infants have been proposed but often show conflicting results, and the percentage of infants experiencing severe COVID-19 varies widely across different studies [1,3,9].

In infants affected by COVID-19, fever emerges as the most common symptom and is one of the most frequent reasons for consultation in the pediatric emergency department (PED) [2,10].

Fever without a clear source may be the sole presenting symptom, and in patients aged 90 days or younger, physical examination and laboratory tests may not be consistently differentiate between viral infections, including SARS-CoV-2, and Severe Bacterial Infection (SBI) [11]. 

Several algorithms have been proposed to identify subgroups at higher risk of SBI, constituting up to 12% of febrile children aged 90 days or younger [11,12,13,14,15]. Among these, the American Academy of Pediatrics guidelines and the “Step-by-Step” approach are the most commonly used, demonstrating a lower incidence of undiagnosed SBI, compared to other scoring systems [13,14]. However, these approaches have been reported to be less sensitive in children with a short duration of fever, often leading to hospitalization and a comprehensive sepsis workup [10,11,14,16]. Moreover, these algorithms do not include the use of rapid diagnostic tests for viral infections, including COVID-19 [11,13,14].

The primary aim of our study was to describe the clinical and biochemical characteristics and outcomes of neonates and young infants with COVID-19 to evaluate disease severity and the need for diagnostic investigations and hospitalization.

Simultaneously, we sought to propose a management pathway for neonates and young infants with COVID-19 in PEDs.

## 2. Materials and Methods

We conducted a retrospective analysis by collecting data from the medical records of infants admitted to the PED of a tertiary children’s hospital (IRCCS Istituto Giannina Gaslini, Genoa, Italy) spanning from 1 November 2020 to 31 October 2023.

We included all patients aged ≤90 days with a confirmed positive antigenic nasal swab for SARS-CoV-2, or detection of SARS-CoV-2 RNA using a real-time quantitative reverse transcriptase polymerase chain reaction (RTq-PCR) from either a nasal swab or saliva sample.

Patients accessing our PED before 1 November 2020 were excluded, as comprehensive screening for SARS-CoV-2 had not yet been implemented for all admitted patients during that period.

Demographic data such as age, sex, presence of older siblings, comorbidities, and perinatal history, along with symptoms on admission, administered therapies, and results of blood and urine tests were collected. Outcome data, length of hospitalization, admission to the neonatal (NICU) or pediatric intensive care unit (PICU), need for invasive or non-invasive ventilation, occurrence of complications, and mortality were also documented.

Patients were categorized into two groups based on age: those younger than 28 days (referred to as the “neonates group”) and those 28 days or older (referred to as the “infants group”). Subsequently, patients were further stratified into febrile and afebrile groups. Among febrile patients, a subdivision was made based on the time elapsed between the onset of fever and admission to the PED, using six hours of fever as the cut-off. A flowchart of the inclusion and exclusion criteria is shown in Figure 1.

The study did not seek local ethics committee approval, as it falls within the purview of the institute’s ethics committee, which had previously approved COVID-19-related data collection (Regione Liguria Ethical Board; IRB#370/2020).

### Statistical Analysis

Mean and standard deviation (SD) were utilized for normally distributed variables, while median and interquartile range (IQR) were represented for non-normally distributed variables. Categorical variables were expressed as numbers and percentages. Group differences were assessed using Kruskal–Wallis test or Mann–Whitney U-test for continuous variables, and chi-square or Fisher’s exact test for categorical variables. Statistical significance was established at *p* < 0.05, and all values were determined using two-tailed tests. Multivariable logistic or linear regression analysis, based on variable types, was performed to identify clinical and laboratory differences between non-hospitalized and hospitalized patients, as well as factors influencing the length of stay. Only variables that demonstrated statistical significance at the univariable level were considered for multivariable models. All statistical analyses were conducted using IBM SPSS Statistics for Windows Version 21.0 (IBM Corp., Armonk, NY, USA).

## 3. Results

### Demographic and Clinical Findings

A total of 274 outpatients were included in the study, reflecting overall incidence of 0.28 SARS-CoV-2 infections per 10,000 PED admissions. The median age was 47 days (IQR 31–68), with 45% being female. SARS-CoV-2 infection was diagnosed in 60 neonates and 214 infants, and no season distribution was observed SARS-CoV-2 infection was detected in 9% of patients aged ≤90 days. The demographic and clinical data are summarized in Table 1. A total of 43.5% patients had at least one older sibling with equal distribution between the two age groups (46.5% of neonates and 42% of infants). However, data were missing for 99 patients due to incomplete medical history.

We observed a higher hospitalization rate in the newborn group compared to the infant group (83.5% vs. 67.5%, *p* = 0.018), although there were no significant differences in the length of hospital stay (LOS).

Twenty-three children (8.5%) had comorbidities, including premature birth, with three premature patients in the newborn group and four in the infant group.

While there was a trend towards a longer length of stay for patients with comorbidities (72 h, IQR 48–96, vs. 48 h, IQR 48–72) this difference did not reach statistical significance (*p* = 0.09). However, patients with comorbidities did not show a higher hospitalization rate (*p* = 0.2).

Seventeen patients (6.2%) had a history of positive maternal vaginal swabs for Group B *Streptococcus* (GBS). In eleven cases, complete intrapartum antibiotic prophylaxis was administered, while the remaining six patients underwent blood tests at birth and in the days following birth. None of the patients presented early neonatal sepsis. Table 1 shows the main symptoms upon admission to the PED. Nineteen patients (7%) were completely asymptomatic, evenly distributed between the neonates and infants groups. Eight infants (3%) were noted to have poor clinical conditions. Fever was the main presenting symptom in 206 patients (76%), occurring as an isolated symptom in 39% of cases, with no significant difference between the two groups (*p* = 0.34). Neonates, however, showed a statistically significant increase in the prevalence of poor feeding (*p* = 0.047) and a shorter time interval between the onset of fever and PED arrival (*p* = 0.004).

Blood and urine tests were performed in 206 (75%) and 192 (70%) patients, respectively. The results are summarized in Table 1, Table 2, Table 3, Table 4 and Table 5.

We did not find any positive blood cultures in the 47 patients tested. Nine patients had a positive urine dipstick for nitrites and/or leukocytes, while only two infants had positive urine cultures, one for *Klebsiella pneumoniae* and one for *Escherichia coli*. Five patients, all from the infant group, presented a coinfection with respiratory syncytial virus (RSV). Lumbar puncture was performed in only one newborn due to fever without a source and poor clinical conditions.

All 18 patients with gastrointestinal symptoms tested negative for bacterial cultures and viral fecal antigens.

Regarding therapy, empiric antibiotics were administered to nine patients (3%) with a pathological urine dipstick and discontinued in seven patients when a negative urine culture was obtained. The remaining two patients with documented urinary tract infections (UTI) continued antibiotic therapy based on antibiotic susceptibility tests. Antiviral drugs were not administered due to the overall favorable clinical course.

The hospitalization rate was 71% (195/274), with 83% in infants aged <29 days, and 67% in infants aged ≥29 days. The median LOS was 48 h in both age groups (Table 1). No patients required hospitalization in the PICU or NICU; only one patient required respiratory support with high-flow nasal cannula (HFNC). Ten patients required low-flow oxygen supplementation upon admission, but none required supplementation during the hospital stay. No secondary SBI were observed, and all hospitalized patients were discharged without evidence of short-term sequelae. No deaths occurred, and none of the patients returned to PED within a month of discharge. Additionally, no cases of Multisystem Inflammatory Syndrome (MIS-C) were registered among the patients included in the study. Differences between hospitalized and non-hospitalized patients are summarized in Table 6. In multivariable analysis, independent factors for hospitalization were fever (*p* = 0.002), poor feeding (*p* = 0.015), and lower age (*p* < 0.001). Table 7 reports possible explanatory variables involved in prolonging LOS: only procalcitonin (PCT) higher than 0.5 ng/dl at arrival remained significant (*p* = 0.036) in the multivariable model.

## 4. Discussion

This study provides clinical insights into one of the largest series of children aged ≤ 90 days admitted to a tertiary PED with a confirmed SARS-CoV-2 infection. Our findings suggest that these patients commonly experience mild or asymptomatic course of the disease, with moderate-to-severe respiratory symptoms such as respiratory distress and apnea, occurring in only a minority of patients (6%), according to previously published reports [3,17,18]. Furthermore, the patients follow a benign clinical course, with unremarkable inflammatory blood test results, often not requiring hospital admission or specific therapies.

### 4.1. Clinical Presentation

Currently, a significant knowledge gap persists regarding the manifestation of the disease, clinical course, and risk factors for severe disease in neonates and infants infected with SARS-CoV-2 [2,17]. Some authors have suggested a potentially higher risk of severe disease in this age group compared to older children [3,4]. However, most of these studies only considered data on hospitalized infants, leaving the complete disease spectrum in this age group only partially explored [17].

Within our series, fever emerged as the most common presenting symptom of COVID-19 in infants (76%), followed by upper respiratory tract (URT) symptoms (34%) and poor feeding (22%) [3,9,15,16,17,18]. In contrast to other reports, we observed a lower frequency of gastrointestinal symptoms. Although it is often associated with fever or URT symptoms, poor feeding can be the only symptom observed. Thus, COVID-19 should also be considered in infants with inadequate milk intake [9,17,19].

We reported a non-negligible percentage (7%) of asymptomatic patients who accessed PED for reasons other than infectious symptoms such as trauma or parental concern due to contact with COVID-19, reinforcing the notion of a benign clinical course. In total, 9% of the patients in our series had comorbidities, with an equal distribution between the two age groups. Although comorbidities may influence the decision to hospitalize, especially in neonates, their disease course was unremarkable, as observed in other series [20,21].

### 4.2. Blood and Urine Test Findings

We found nine pathological urine dipsticks, and among these, we confirmed only two cases of UTIs with positive urine cultures, representing 1% of all febrile patients, consistent with a previously reported case series [22]. Seven out of nine patients with false-positive urine dipsticks received unnecessary antibiotic treatment. Based on these data and previous studies, it is advisable to consider UTIs even in febrile patients with a positive SARS-CoV-2 swab and without other clinical symptoms [14,23]. However, maintaining a high clinical suspicion is crucial to avoid misdiagnosis of UTIs, depending on the urine collection method, and unnecessary treatments [14,24,25].

We observed a higher percentage of blood tests performed in febrile newborns compared to infants (100% vs. 87%, *p* = 0.013), consistent with main recommendations advocating for more cautious clinical management in those under 28 days of life [13,14].

In addition to absolute values, we considered the cut-off values suggested for white blood cells (WBC), absolute neutrophil count (ANC), C-reactive protein (CRP), and PCT by previous authors [10,13,14,16].

However, blood test findings were largely inconsequential, showing no significant differences in absolute values across various groups, except for a slightly higher WBC count in neonates with more than six hours of fever.

Lower WBC and ANC values were detected in the infant group, both within and after six hours of fever (Table 2 and Table 3). This can be partly explained by physiologically lower ANC values beyond neonatal age [18]. Furthermore, this was not associated with a worse prognosis.

No significant differences were found when comparing febrile and afebrile patients, suggesting a mild inflammatory state in COVID-19 patients and supporting a benign clinical course (Table 4 and Table 5).

Uniquely, our study utilized cut-off values of the main inflammation indices suggested by the AAP and Step-by-step in a COVID-19 case series [13,14,16]. Considering the pathological values of CRP > 2 mg/dL, we observed worse values in infants with fever within six hours, suggesting a non-progression of the inflammatory state in the hours following fever (Table 2 and Table 3).

Given the known high prevalence of SBI in children aged ≤90 days and the understanding that inflammatory markers performed within 12 h of fever may not differentiate between common benign viral infections and SBI, it is noteworthy that we did not detect any secondary SBI [10,13,14]. In light of our results and literature data, we suggest that laboratory tests in COVID-19 infants should not be routinely performed except in selected cases (poor clinical condition, poor feeding, comorbidities, and clinical concerns) [11]. Finally, the tests should not be repeated if the initial results fall within the normal range.

### 4.3. Management and Outcome

Five infants presented a coinfection with RSV and required oxygen supplementation, and one patient developed acute bronchiolitis with respiratory distress, requiring respiratory support with HFNC. This highlights the importance of considering alternative diagnoses beyond COVID-19, especially in cases of respiratory failure or the need for respiratory support, and the possibility of coinfections with more aggressive respiratory viruses [19,26].

Given the absence of specific guidelines, our standard practice was to hospitalize most positive infants and newborns for observation, following other guidelines for managing fever in patients aged ≤90 days [1,2,10,14]. This inclination is underlined by our multivariable analysis, where fever and lower age, along with poor feeding, emerged as independent factors influencing hospitalization. This is further confirmed by the higher hospitalization rate in neonates (*p* = 0.018), despite presenting a benign clinical course without complications, with an average length of hospital stay comparable to the infant group (*p* = 0.94) [3,17].

All our patients rapidly improved and were discharged, suggesting a favorable outcome of community-acquired COVID-19 in newborns and infants [3,6,27]. A PCT higher than 0.5 ng/mL upon admission was the only factor correlated with a longer LOS, potentially reflecting a more precautionary approach in patients with positivity of this inflammation marker that is universally correlated with sepsis [13,14].

The majority of hospitalized children were healthy, with no underlying clinical conditions, and were admitted primarily due to concerns related their young age, as reported in other case series [2,3,6,17,18,19,27,28].

Our data suggest that children <90 days old with COVID-19 are in good health, do not require hospitalization or specific care, and can, therefore, be easily managed at home [1,28,29]. Moreover, COVID-19 in infants under 90 days of age is milder than other acute viral illnesses in the same age group regarding the risk of developing bronchiolitis and/or the need for oxygen supplementation [19].

Therefore, we believe that a patient in good clinical condition with a positive swab for SARS-CoV-2 could be safely discharged without undergoing a full sepsis work-up [2,11,18]. In agreement with previous studies, we recommend that hospitalization should be considered in conditions that may lead to a more complicated clinical course or suboptimal home management, such as low-income family compliance or concern, low respiratory tract involvement, oxygen supplementation at admission, RSV coinfection, or poor feeding [17,18,19,30,31].

Additionally, hospitalization should also be recommended in case of underlying comorbidities or in case of prematurity <37 weeks of gestation, as these patients may be at a higher risk of severe respiratory disease, and a greater need for ICU admission [3,17,18,20,21,32,33].

Similarly, hospitalization should be advised for unwell patients at the time of admission due to reported anecdotal cases of myocarditis and encephalitis related to SARS-CoV-2, even in the neonatal and childhood periods [2,34].

Figure 2 summarizes the suggestions provided in the text.

## 5. Limitations

Our study’s retrospective and monocentric design may introduce bias, as our findings may not be generalizable to other settings. Missing data for certain demographic items can impact sample representativeness and reduce the statistical power of our analysis, introducing bias in parameter estimation. Furthermore, the absence of complete data on long-term follow-up beyond 30 days after discharge prevents us from drawing conclusions regarding long-term COVID-19 complications.

## 6. Conclusions

The lack of information on COVID-19 characteristics in neonates and infants aged <90 days remains a challenge for PED physicians. In our study, these patients experienced a non-specific, milder, or asymptomatic form of COVID-19. In addition, blood tests were consistent with a mild inflammatory response, emphasizing the need for such tests only in children presenting with noticeable illness [5,27].

According to our findings, precautionary hospitalization of infants who do not display signs of illness or without comorbidities may not be justified. Infants under 90 days of age could be treated like those with any other viral infections and be discharged to a reliable family with appropriate follow-up. Further larger prospective studies are necessary to confirm our findings and design a unique clinical management strategy for newborns and infants with COVID-19.

Finally, our research suggests that further studies are needed to integrate common algorithms for febrile infants with rapid viral infection diagnostic tests to reduce unnecessary hospitalizations.

## Figures and Tables

**Figure 1 healthcare-12-00528-f001:**
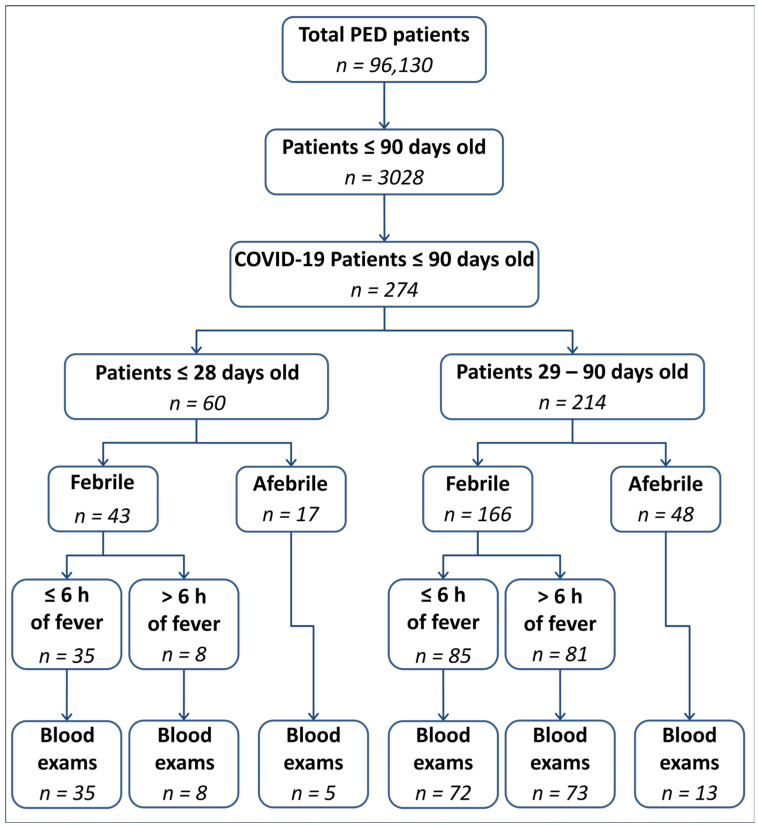
Flowchart of the inclusion and exclusion criteria. PED = Pediatric Emergency Department; COVID-19 = CoronaVirus Disease 19.

**Figure 2 healthcare-12-00528-f002:**
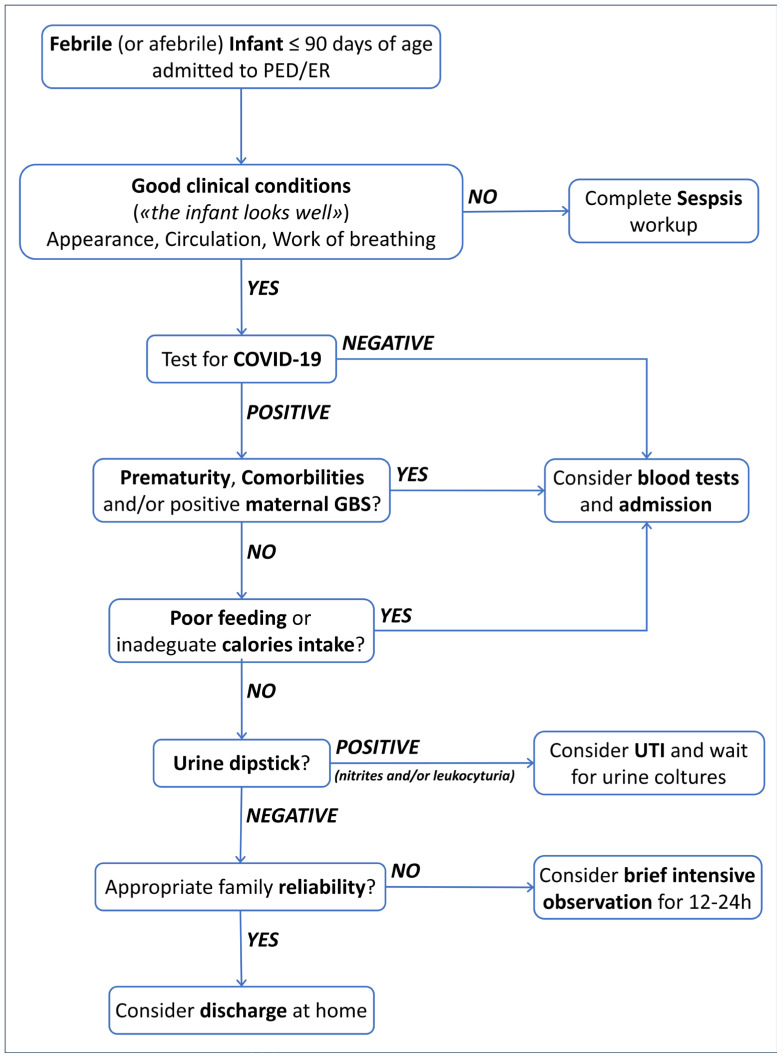
Shows our proposed management pathway, summarizing the previous suggestions.

**Table 1 healthcare-12-00528-t001:** Demographic, clinical, and prognostic findings of enrolled patients. IQR = interquartile range; LOS = length of stay; RSV = respiratory syncytial virus; GI = gastro-intestinal; LRT = lower respiratory tract; URT = upper respiratory tract; HFNC = high flow nasal cannula.

	Total *n* = 274	Patients <29 Days Old *n* = 60	Patients ≥29 Day Old *n* = 214	*p*
Age, median (IQR 25–75)	47 (31–68)	16 (10.75–22.25)	54.5 (41–71)	<0.0001
Sex, male (%)	152 (55.0%)	33 (55%)	119 (55.5%)	0.93
Comorbidities, yes (%)	23 (9%)	5 (8.5%)	18 (9%)	0.81
Older sibling, yes (%)	76 (43.5%)	22 (47%)	54 (42%)	0.58
Maternal GBS, positive (%)	17 (6.5%)	6 (10.5%)	11 (5.5%)	0.19
Admission, yes (%)	195 (71%)	50 (83.5%)	145 (67.5%)	0.018
LOS in h, median (IQR 25–75)	48 (48–72)	48 (48–78)	48 (48–72)	0.94
Poor clinical condition, yes (%)	8 (3%)	3 (5%)	5 (2.5%)	0.27
Asymptomatic, yes (%)	19 (7%)	6 (10%)	13 (6%)	0.29
Fever, yes (%)	209 (76%)	43 (71.5%)	166 (77.5%)	0.34
Fever < 6 h, yes (%)	120 (57.5%)	35 (81%)	85 (51%)	0.004
Blood exams, yes (%)	206 (75%)	48 (80%)	158 (74%)	0.32
Blood exams in fever, yes (%)	188 (90%)	43 (100%)	145 (87%)	0.013
Blood exams in afebrile, yes (%)	65 (24%)	5 (29.5%)	13 (27%)	0.85
Urine exams, yes (%)	192 (70%)	38 (63%)	154 (78%)	0.19
Positive nitrites, yes (%)	2 (1%)	0 (%)	2 (1.5%)	0.47
Leukocyturia, yes (%)	9 (4.5%)	0 (%)	9 (6%)	0.12
Urine culture collected, yes (%)	66 (34%)	12 (31.5%)	54 (35%)	0.68
Urine culture, positive (%)	2 (3%)	0 (0%)	2 (3.5%)	0.49
Blood culture collected, yes (%)	47 (23%)	11 (23%)	36 (22.5%)	0.98
Blood culture, positive (%)	0 (0%)	0 (%)	0 (%)	/
RSV coinfection, yes (%)	5 (2%)	0 (0%)	5 (2.5%)	0.23
GI symptoms, yes (%)	18 (6.5%)	3 (5%)	15 (7%)	0.57
Poor feeding, yes (%)	61 (22%)	19 (31.5%)	42 (19.5%)	0.047
LRT symptoms, yes (%)	19 (7%)	2 (3.5%)	17 (8%)	0.21
URT symptoms, yes (%)	93 (34%)	21 (35%)	72 (33.5%)	0.84
Dyspnea, yes (%)	16 (6%)	3 (3.5%)	13 (6%)	0.69
Apnea, yes (%)	6 (2%)	1 (1.5%)	5 (2.5%)	0.75
Cutaneous rash, yes (%)	7 (2.5%)	1 (1.5%)	6 (3%)	0.62
Oxygen supplementation, yes (%)	10 (3.5%)	3 (3.5%)	7 (3.5%)	0.52
HFNC, yes (%)	1 (0.5%)	0 (0%)	1 (0.5%)	0.59

**Table 2 healthcare-12-00528-t002:** Comparison of inflammatory marker values between the two age groups within six hours of fever onset. Continuous variables are described as median and interquartile range (IQR), and categorical variables as absolute and relative frequencies. CRP = C-reactive protein; PCT = procalcitonin; WBC = white blood cells; ANC = absolute neutrophils count.

	Total Patients *n* = 107	Patients <29 Days Old *n* = 35	Patients ≥29 Days Old *n* = 72	*p*
CRP in mg/dL, median (IQR 25–75)	0.00 (0.00–0.00)	0.00 (0.00–0.00)	0.00 (0.00–0.08)	0.34
CRP > 2 mg/dL, yes (%)	11 (10%)	0 (0%)	11 (15%)	0.014
PCT in ng/mL, median (IQR 25–75)	0.11 (0.05–0.14)	0.11 (0.08–0.13)	0.10 (0.05–0.14)	0.85
PCT > 0.5 ng/mL, yes (%)	2 (2%)	1 (3%)	1 (1.5%)	0.59
WBC in cells/mm^3^, median (IQR 25–75)	7100 (5300–8750)	7900 (6000–8850)	6650 (5150–8375)	0.16
WBC > 10,000 cells/mm^3^, yes (%)	12 (11%)	3 (8.5%)	9 (12.5%)	0.54
ANC in cells/mm^3^, median (IQR 25–75)	2300 (1750–3400)	2800 (2100–3600)	2150 (1500–3200)	0.03
ANC < 1000 cells/mm^3^, yes (%)	3 (3%)	1 (3%)	2 (3%)	0.98
ANC > 5000 cells/mm^3^, yes (%)	6 (6%)	0 (0%)	6 (8.5%)	0.07

**Table 3 healthcare-12-00528-t003:** Comparison of inflammatory marker values between the two age groups more than six hours after fever onset. Continuous variables are described as median and interquartile range (IQR), and categorical variables as absolute and relative frequencies. CRP = C-reactive protein; PCT = procalcitonin; WBC = white blood cells; ANC = absolute neutrophils count.

	Total Patients *n* = 81	Patients <29 Days Old *n* = 8	Patients ≥29 Days Old *n* = 73	*p*
CRP in mg/dL, median (IQR 25–75)	0.00 (0.00–0.00)	0.00 (0.00–0.04)	0.00 (0.00–0.00)	0.51
CRP > 2 mg/dL, yes (%)	2 (2.5%)	1 (12.5%)	1 (1.5%)	0.054
PCT in ng/mL, median (IQR 25–75)	0.11 (0.05–0.16)	0.15 (0.10–0.19)	0.10 (0.05–0.15)	0.65
PCT > 0.5 ng/mL, yes (%)	1 (1%)	0 (0%)	1 (1.5%)	0.73
WBC in cells/mm^3^, median (IQR 25–75)	7200 (5150–9800)	10,200 (7400–11,000)	6500 (4300–8900)	0.005
WBC > 10,000 cells/mm^3^, yes (%)	19 (23.5%)	5 (62.5%)	14 (19%)	0.006
ANC in cells/mm^3^, median (IQR 25–75)	1970 (1000–2525)	2100 (1600–2550)	1600 (1000–2500)	0.25
ANC < 1000 cells/mm^3^, yes (%)	18 (22%)	2 (25%)	16 (22%)	0.84
ANC > 5000 cells/mm^3^, yes (%)	5 (6%)	1 (12.5%)	4 (5.5%)	0.43

**Table 4 healthcare-12-00528-t004:** Comparison of age, sex, and values of inflammatory markers in patients aged ≤28 days with or without fever at the time of PED admission. Continuous variables are described as median and interquartile range (IQR), and categorical variables as absolute and relative frequencies. CRP = C-reactive protein; PCT = procalcitonin; WBC = white blood cells; ANC = absolute neutrophil count.

	Total < 29 Days Old Patients *n* = 48	Febrile Patients *n* = 43	Afebrile Patients *n* = 5	*p*
Age, median (IQR 25–75)	16 (10.75–22.25)	16 (10.50–22.00)	16 (11.00–23.00)	0.67
Sex, male (%)	25 (52%)	22 (51%)	3 (60%)	0.70
CRP in mg/dL, median (IQR 25–75)	0.00 (0.00–0.00)	0.00 (0.00–0.00)	0.00 (0.00–0.00)	0.60
CRP > 2 mg/dL, yes (%)	1 (2%)	1 (2.5%)	0 (0%)	0.73
PCT in ng/mL, median (IQR 25–75)	0.11 (0.07–0.15)	0.12 (0.08–0.15)	0.08 (0.04–0.12)	0.87
PCT > 0.5 ng/mL, yes (%)	1 (2%)	1 (2.5%)	0 (0%)	0.73
WBC in cells/mm^3^, median (IQR 25–75)	7900 (6300–9200)	7950 (6400–9975)	8500 (6750–10,400)	0.20
WBC > 10,000 cells/mm^3^, yes (%)	9 (19%)	8 (18.5%)	1 (20%)	0.94
ANC in cells/mm^3^, median (IQR 25–75)	2500 (1950–3500)	2450 (2000–3475)	1500 (1450–2150)	0.065
ANC < 1000 cells/mm^3^, yes (%)	1 (2%)	3 (7%)	0 (0%)	0.54
ANC > 5000 cells/mm^3^, yes (%)	0 (0%)	1 (2.5%)	0 (0%)	0.73

**Table 5 healthcare-12-00528-t005:** Comparison of age, sex, and values of inflammatory markers in patients aged 29–90 days with or without fever at the time of PED admission. Continuous variables are described as median and interquartile range (IQR), and categorical variables as absolute and relative frequencies. CRP = C-reactive protein; PCT = procalcitonin; WBC = white blood cells; ANC = absolute neutrophil count.

	Total 29–90 Days Old Patients *n* = 158	Febrile Patients *n* = 145	Afebrile Patients *n* = 13	*p*
Age, median (IQR 25–75)	54.5 (41–71)	55.5 (42–72)	51.00 (37.75–67.75)	0.26
Sex, male (%)	88 (55.5%)	81 (56%)	7 (54%)	0.88
CRP in mg/dL, median (IQR 25–75)	0.00 (0.00–0.00)	0.00 (0.00–0.00)	0.00 (0.00–0.00)	0.27
CRP > 2 mg/dL, yes (%)	12 (7.5%)	12 (8%)	0 (0%)	0.28
PCT in ng/mL, median (IQR 25–75)	0.09 (0.03–0.14)	0.10 (0.05–0.15)	0.00 (0.00–0.05)	0.32
PCT > 0.5 ng/mL, yes (%)	2 (1.5%)	2 (1.5%)	0 (0%)	0.66
WBC in cells/mm^3^, median (IQR 25–75)	6600 (4800–8900)	6600 (4800–8900)	8900 (5900–9550)	0.19
WBC > 10,000 cells/mm^3^, yes (%)	25(16%)	23 (16%)	2(15%)	0.96
ANC in cells/mm^3^, median (IQR 25–75)	2000 (1285–2800)	2000 (1200–2700)	1700 (1400–2500)	0.59
ANC < 1000 cells/mm^3^, yes (%)	20 (12.5%)	18 (12.5%)	2 (15%)	0.75
ANC > 5000 cells/mm^3^, yes (%)	10 (6%)	10 (7%)	0 (0%)	0.32

**Table 6 healthcare-12-00528-t006:** Univariate and multivariate analysis of independent factor assessed at the time of PED presentation in hospitalized versus non-hospitalized patients. Categorical variables are described as absolute and relative frequencies. GBS = group B Streptococcus; LRT = lower respiratory tract; URT = upper respiratory tract; WBC = white blood cells; ANC = absolute neutrophil count; CRP = C-reactive protein; PCT = procalcitonin; * Haldane-Anscombe correction.

	Hospitalization		Univariable		Multivariable	
	No	Yes	*p*	Odds Ratio (95% CI)	*p*	Odds Ratio (95% CI)
Sex			0.893	1.06 (0.628–1.79)		
Female	36 (29.5%)	86 (70.5%)				
Male	43 (28.3%)	109 (71.7%)				
Maternal positive GBS	3 (17.6%)	14 (82.4%)	0.572	1.79 (0.499–6.45)		
Comorbidities	4 (17.4%)	19 (82.6%)	0.457	1.68 (0.548–5.13)		
Older sibling	11 (14.5%)	65 (85.5%)	0.546	1.31 (0.579–2.98)		
Poor clinical condition	0 (0)	8 (100%)	0.11	7.12 (0.406–125) *		
Fever	54 (25.8%)	155 (74.2%)	0.06	1.79 (0.997–3.23)	0.002	2.79 (1.44–5.39)
Dyspnea	0 (0)	16 (100%)	0.008	14.6 (0.866–247) *	0.986	/
URT symptoms	24 (25.8%)	69 (74.2%)	0.482	1.25 (0.715–2.2)		
LRT symptoms	1 (5.3%)	18 (94.7%)	0.017	7.93 (1.04–60.5)	0.128	5.43 (0.61–47.98)
Vomiting	0 (0)	5 (100%)	0.326	4.59 (0.251–84.0) *		
Diarrhea	7 (50%)	7 (50%)	0.125	0.383 (0.130–1.13)		
Poor feeding	9 (15%)	51 (85%)	0.009	2.75 (1.28–5.91)	0.015	2.78 (1.21–6.35)
Cutaneous rash	1 (14.3%)	6 (85.7%)	0.677	2.48 (0.293–20.9)		
WBC > 10,000/mmc	0 (0)	15 (100%)	0.213	5.69 (0.323–100) *		
ANC > 5000/mmc	1 (16.7%)	5 (83.3%)	0.58	0.745 (0.08–6.85)		
CRP > 2 mg/dL	0 (0)	1 (100%)	1	1.10 (0.05–22.4) *		
PCT > 0.5 ng/mL	0 (0)	2 (100%)	1	1.04 (0.04–22.9) *		
Age (days, median)	57	42	<0.001	0.98 (0.968–0.992)	<0.001	0.976 (0.964–0.989)

**Table 7 healthcare-12-00528-t007:** Univariate and multivariate analysis of independent factor assessed at the time of PED presentation in different LOS. LOS = length of stay; GBS = group B Streptococcus; LRT = lower respiratory tract; URT = upper respiratory tract; WBC = white blood cells; ANC = absolute neutrophil count; CRP = C-reactive protein; PCT = procalcitonin.

	LOS (h, Median)	Univariable		Multivariable	
		*p*	Estimate (95% CI)	*p*	Estimate (95% CI)
Sex (male)	48	0.183	4.48 (−14.8–2.85)		
Maternal positive GBS	84	0.020	20.3 (3.29–37.3)	0.154	19.10 (−7.35–45.6)
Comorbidities	72	0.051	14.7 (−0.06–29.4)		
Older sibling	48	0.246	6 (−4.18–16.2)		
Poor clinical condition	108	<0.001	46.2 (25.2–67.3)	0.190	25.14 (−12.77–63.1)
Fever	48	0.629	−2.69 (−13.6–8.26)		
Dyspnea	84	0.003	23.8 (8.21–39.4)	0.979	−0.443 (−34.22–33.3)
URT symptoms	72	0.185	6.18 (−2.97–15.3)		
LRT symptoms	96	<0.001	26.1 (11.4–40.7)	0.139	28.47 (−9.5–66.4)
Vomiting	72	0.338	13.4 (−14.2–41)		
Diarrhea	48	0.441	−9.19 (−32.7–14.3)		
Poor feeding	48	0.006	13.8 (3.99–23.7)		
Cutaneous rash	60	0.854	2.37 (−23–27.7)		
WBC > 10,000/mmc	72	0.210	11.3 (−6.46–29)		
ANC > 5000/mmc	48	0.249	−17 (−46–12.1)		
CRP > 2 mg/dL	96	0.183	24.7 (−11.9–61.4)		
PCT > 0.5 ng/mL	120	0.012	58.2 (13.3–103.1)	0.020	54.64 (9.07–100.2)

## Data Availability

Data are contained within the article.

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
