# Peer review of "Clinical Course, Laboratory Findings, and Prognosis of SARS-CoV-2 Infection in Infants up to 90 Days of Age: A Single-Center Experience and a Proposal for a Management Pathway"

_healthcare, 2024, doi:10.3390/healthcare12050528_

Round 1
Reviewer 1 Report
Comments and Suggestions for Authors
General comments
Thank you for inviting me to review the article “Clinical course, laboratory findings and prognosis of SARS-CoV-2 infection in infants up to 90 days of age: a single center experience and a proposal for a management pathway.” The article addresses a topic of interest, shedding light on aspects that have received little attention in the literature. I have some suggestions to improve the manuscript's quality and clarity.
Abstract
Integrate more numerical data, such as the number of newborns in each age group, specific symptoms observed, or median hospitalization duration.
Explain the importance of the SARS-CoV-2 infection rate in relation to PED admissions.
In summarizing the important findings and their consequences, the conclusion should be more explicit.
Introduction
SARS-CoV-2 is an acronym that stands for “Severe Acute Respiratory Syndrome Coronavirus 2,” not “Acquired.”
More information on the contradicting statistics about pediatric patients, particularly those under 90 days old, is needed. What are the contradictory findings, and what areas require more research?
Consider a smoother transition from your study's background to its precise aims. You may want to state the research questions or hypotheses directly.
Materials and Methods
Explain why the date range of November 1, 2020 to October 31, 2023 was chosen. Was it due to changes in screening procedures, data availability, or something else?
Consider including more information about the comorbidity and perinatal history criteria. What comorbidities were considered, and how were they diagnosed or defined? Furthermore, why didn't you evaluate if the participants were full-term neonates or preterm newborns? This could aid in understanding some of the factors behind the severity of COVID-19 symptoms.
Insert Figure 1 after it is mentioned. Include a short textual description of the flowchart (Figure 1) in the main paragraph as well.
Results
Identify the missing data (99 charts) and discuss briefly how this may affect the interpretation of the results. It is preferable to discuss any limitations openly.
Again, offer further information about the sorts of comorbidities seen, particularly preterm condition. Indicate whether preterm was considered alone or if other comorbidities were present.
Why is Table 1 captioned twice and differently? It's confusing. Separate “Older” from “Sibling” as well.
Insert Tables 2 and 3 after they have been discussed so that the reader understands what you are commenting on.
Tables 4 and 5 should also be discussed in this section.
Discussion
Begin by summarizing the important findings from the results section. This gives readers a fast summary before digging into the detailed analysis.
The issues you raised deserve to be evaluated in sections. To improve clarity and make it easier for readers to follow your ideas, divide the discussion into subsections. As an example, you may include sections on clinical presentation, test findings, and management recommendations.
Furthermore, clearly define your clinical management suggestions based on the study findings. Discuss the hospitalization criteria and the significance of elements such as family compliance, respiratory tract involvement, and comorbidities.
Finish the debate by identifying potential areas for future investigation. Identify specific gaps in knowledge that your study uncovered and suggest future research directions.
Finally, there is no mention of the Ethics Committee, nor is there any mention of Informed Consent. This is a significant limitation that may prevent the paper from being published.
Reviewer 2 Report
Comments and Suggestions for Authors
I appreciate your submission and the effort you've put into your work. However, there are several changes that must be addressed before considering this manuscript as a potential candidate for publication.
Abstract:
Review the structure of the abstract. Lines 22 and 23 are irrelevant and disrupt the flow of reading.
Clarify the meaning of "Blood tests were 28 unremarkable."
Present data in percentages rather than absolute numbers (e.g., "Two infants had urinary tract infections with positive 29 urine cultures; five patients had an RSV coinfection").
Introduction:
Omit lines 42 to 44 as this information is already part of common knowledge and does not contribute to the discussion or understanding of the problem.
Methods:
Clarify the acronyms NICU and PICU.
There is no mention of ethical committee approval anywhere. Please provide the approval document.
Results:
Improve the clarity of the wording in line 110.
Consider having the manuscript reviewed by a colleague proficient in English.
Clarify the trend referred to in line 116. Did you conduct a p trend or a jointpoint analysis to assess this trend? Why is it not reported?
Remember, numbers from 1 to 10 should be written in text, starting from 11 in numerical form (e.g., line 119).
Table 1 lacks a title and has the footnote in its place.
Discussion:
Many elements in the discussion correspond to the results section.
Conduct a more critical analysis of the findings instead of merely reporting cases and results.
Given the abundance of data, consider performing multivariate analysis to enrich the study.
Clearly state the manuscript's contribution and what knowledge gaps it addresses.
Finally, the manuscript has significant issues with English language proficiency. I recommend a thorough language review.
Best regards,
Comments on the Quality of English Languagenumbers from 1 to 10 should be written in text, starting from 11 in numerical form (e.g., line 119).
Round 2
Reviewer 1 Report
Comments and Suggestions for Authors
The authors have made revisions to the manuscript based on my comments and suggestions. I believe that now the paper can be accepted in its current form.
Author Response
We deeply thank the reviewer for his helpful suggestions.
Reviewer 2 Report
Comments and Suggestions for Authors
I hope this message finds you well. I have completed the review of the manuscript with the title "Clinical course, laboratory findings, and prognosis of SARS-CoV-2 infection in infants up to 90 days of age: A single-center experience and a proposal for a management pathway," and I would like to thank the authors for their meticulous work and insightful contribution to the field.
I am pleased to note that the authors have graciously accepted the proposed changes, and I believe the manuscript is now in good shape for publication. However, I have one specific request for revision related to Tables 6 and 7.
I kindly suggest that the authors include the confidence intervals for both univariate and multivariate models in these tables. This addition would enhance the statistical robustness of the findings and provide readers with a more comprehensive understanding of the results. Additionally, I recommend a thorough review of the font type used in these tables to ensure consistency and readability.
Secondly, I recommend ensuring the manuscript includes explicit information regarding the ethical approval granted by the relevant committee for the conducted research. This information is pivotal for maintaining transparency and upholding ethical standards in scientific research.
Thank you for considering my feedback, and please convey my appreciation to the authors for their valuable contribution.
Author Response
We deeply thank the reviewer for his helpful suggestions. We corrected the table and changed its font, leaving the choice for the best layout to the Editor.
As we have already explained previously, no approval was required from our local ethics committee for this study. However, at the beginning of the pandemic way back in 2020, the ethics committee of our Institute approved any data collection relating to COVID patients, consequently we considered that this study could fall into this type of data collection.
Weadded the sentence indicating the approval number in the materials and methods section.
Please find a R2 manuscript with all requested changes.